# Vitamin D Status for Chinese Children and Adolescents in CNNHS 2016–2017

**DOI:** 10.3390/nu14224928

**Published:** 2022-11-21

**Authors:** Yichun Hu, Shan Jiang, Jiaxi Lu, Zhenyu Yang, Xiaoguang Yang, Lichen Yang

**Affiliations:** Key Laboratory of Trace Element Nutrition of National Health Commission, National Institute for Nutrition and Health, China CDC, Beijing 100050, China

**Keywords:** vitamin D, Chinese children and adolescents, 25-hydroxyviramin D, CNNHS 2016–2017

## Abstract

Vitamin D is very important in maintaining children’s bone health because of its regulatory role in calcium and phosphate metabolism. To better understand vitamin D status and related risk factors of children and adolescents in China, this study analyzed the 25-hydroxyvitamin-D (25(OH)D) concentration of children and adolescents aged 6–17 years in China and assessed the risk factors of vitamin D deficiency and insufficiency. We analyzed the data of 25(OH)D concentration collected from the China National Nutrition and Health Survey of Children and Lactating Mothers in 2016–2017 (CNNHS 2016–2017). The age, sex, region type, ethnicities, season, weight and height were recorded, measured by unified questionnaire, equipment and standards. The concentration of 25(OH)D was detected by LC-MS/MS. A total of 64,391 participants from the cross-sectional study of CNNHS in 2016–2017 were included in this study. The median serum 25(OH)D concentration was 17.70 (13.20–22.68) ng/mL, 18.70 (14.10–23.80) ng/mL in boys and 16.60 (12.40–21.40) ng/mL in girls. The total prevalence rate of vitamin D deficiency and insufficiency was 65.98% when the cut-off was 20 ng/mL, 60.42% for boys and 71.99% for girls. According to the results of logistic regression analysis, girls aged above 12 y, living in midlands and northern regions, in spring and winter seasons and with abdominal obesity will have a significantly increased risk of vitamin D deficiency and insufficiency among Chinese children and adolescents. The results showed that vitamin D deficiency and insufficiency are very common among children and adolescents aged 6–17 y in China. They should be encouraged to have more effective sunlight exposure, increased intake of vitamin D from food or supplements in their diet, especially for those aged above 12 y, living in the northern or midlands areas, in spring and winter, and abdominally obese.

## 1. Introduction

Vitamin D (calciferol) is an essential fat-soluble vitamin in the human body. It has been reported that vitamin D deficiency (VDD) is relative to many diseases, such as fractures, osteoporosis [1], cardiovascular diseases [2], specific cancer [3], and auto-immune diseases [4]. Among them, the relationship between vitamin D and bone health is well known. Vitamin D plays an important role in the metabolism of calcium and phosphorus, so it is crucial for bone growth and bone mineral metabolism. Childhood and adolescence are periods of rapid growth, and they need a variety of nutrients, including vitamin D. Sufficient vitamin D in childhood is essential for normal skeletal and immune growth and development. VDD can lead to rickets, a bone-softening disease that also increases the risk of bone fractures in children, adolescents, and adults. However, several recent reviews of vitamin D status worldwide suggest widespread VDD [5], regardless of a country’s human development index and latitude. VDD has become a major public health problem for all age groups worldwide [1,6]. With that, our group has been monitoring the vitamin D status of the entire population since the Chinese National Nutrition and Health Survey (CNNHS 2010–2012), and it has become a continuous survey indicator of CNNHS.

Serum or plasma 25(OH)D is a recognized indicator for assessing vitamin D nutritional status because of a long half-life (15 days) in the body, and it is relatively stable, sufficient in the blood, and it can respond to the recent endogenous and exogenous (diet or supplement source) vitamin D intake [5]. However, the cut-offs for vitamin D deficiency and insufficiency is still an area of controversy [7]. We adopted the cut-offs recommended by the American Institute of Medicine (IOM) [8] and the European Society for Paediatric Gastroenterology, Hepatology and Nutrition [9], 20 ng/mL and 12 ng/mL, below which is considered as vitamin D insufficiency (VDI) and VDD, respectively.

The aim of this study was to analyze the vitamin D nutritional status and risk factors of VDD of Chinese children and adolescents aged 6–17 years from the China National Nutrition and Health Survey of Children and Lactating Mothers (CNNHS 2016–2017). It is hoped that the results of this study will contribute to the development of policy recommendations for VDD interventions.

## 2. Materials and Methods

### 2.1. Participants and Ethics

The China National Nutrition and Health Survey is a periodic cross-sectional survey of the civilian population of China since 1959 [10]. The survey is divided into two parts, one is the nutritional survey for adults (≥18 years old, including pregnant women) and the other is for children, adolescents and lactating mothers. In the CNNHS 2015–2017, CNNHS 2015–2016 is for adults, and CNNHS 2016–2017 is for children, adolescents and lactating mothers. The National Institute for Nutrition and Health, Chinese Center for Disease Control and Prevention (NINH, China CDC) conducted the survey. All the data of participants aged 6~17 years old were collected from CNNHS 2016–2017. In the survey, NINH adopted a complex, multistage, stratified cluster random sampling method to select all the participants. All the children and adolescents aged 6–17 y were selected from schools in each survey site, half boys and half girls. In this study, after excluding participants without qualified important information, including anthropometric measurement data and laboratory serum 25(OH)D detection, etc., 64,391 children and adolescents were included. All participants and their guardian were given informed consent in writing to participate in the study. The Ethics Review Board of NINH, China CDC approved the protocol (No. 201614).

### 2.2. Data Collection, Body Measurement and Blood Sample Detection

The CNNHS national project workgroup was founded in NINH, China CDC. We designed a unified survey and questionnaire to carry out the investigation survey with unified equipment and methods. The basic information of the participants (including age, sex, region type, ethnicity etc.) was collected by standardized questionnaires. The anthropometric measurements included height and weight and were measured by unified method and equipment, as described by Shi et al. [11]. Body Mass Index (BMI) was calculated by body weight (kg) and height (m). The waist circumference (WC) was measured twice at the midpoint of the line between the lower arch margin of the middle axillary rib and the iliac crest by a soft tape.

A total of 6 mL of fasting venous blood was collected and then centrifuged (1500× *g*, 15 min) 30 min after was blood taken. The upper serum was separated and stored at −20 °C in the laboratory where the survey site was located. All the blood specimens from all the 150 investigation points were transported to the bio samples bank located in NINH, China CDC by cold chain. All the samples were stored at a −70 °C refrigerator before determination, and 25(OH)D was detected by liquid chromatography-mass spectrometry (LC-MS/MS).

### 2.3. Data Collection and Definition

All the information of participants in this study was collected from every investigation point and logged into the systematic platform of the national survey of nutrition and health status for Chinese residents, and region type was recorded and classified accordingly [12,13]. Based on self-report, demographic data (including age, sex, ethnicity and etc.) were recorded. Age groups were classified according to different learning stages, 6–11 y (primary school), 12–14 y (junior high school) and 15–17 y (high school). All ethnicities except Han are classified as minorities. China’s Qinling Mountains and Huaihe River were recognized as the boundary to divide the North and the South. The growth level was defined according to the Chinese screening standards for overweight and obesity and for malnutrition among school-age children and adolescents [14,15]. All the participants were classified as thin, normal, overweight, and obesity according to age-and sex- specific cut-off points of height and/or BMI. Season was recorded according to the date of blood drawn. March, April and May are spring months; June, July and August are summer months; September, October, November are autumn months; December, January and February are winter months. Abdominal obesity is determined by the cut-off points of age- and sex-specific 90th percentile for Chinese children and adolescents [16]. A serum 25(OH)D concentration lower than 12 ng/mL (30 nmol/L) was considered VDD, and considered VDI if it was lower than 20 ng/mL (50 nmol/L) but higher or equal to 12 ng/mL [16]. In this study, VDD and VDI were grouped together as vitamin D inadequacy (<20 ng/mL), while 25(OH)D concentration above and equal to 20 ng/mL was considered vitamin D sufficient.

### 2.4. Data Analyses

All the participants were divided into different sub-groups by different hypothesized predictors for vitamin D status. Serum 25(OH)D concentrations were record by using P50 (P25~P75) because of inconsistency with the normal distribution. The concentration of 25(OH)D in each subgroup was compared by Kruskal–Wallis test. The prevalence of VDD, VDI and vitamin D sufficiency were presented as percentages (%) and the rates of subgroups were compared by chi square test. We utilized multivariable logistic regression analysis to describe the relationship between vitamin D inadequacy and possible predictors (e.g., age, sex, region type, latitude, season, growth level and abdominal obesity). All the data was analyzed by SAS 9.4 software, and the difference was statistically significant with *p* < 0.05.

## 3. Results

### 3.1. Basic Characteristics

The serum 25(OH)D concentration of 64,391 Chinese children and adolescents (32,168 boys and 32,223 girls) were analyzed (Table 1). The median age was 11.4 years old (interquartile range (IQR) 8.9–14.0 y, range 6.0–17.9 y). A total of 35,780 were children aged 6–11.9 y, 15,801 were adolescents aged 12–14.9 y and 12,810 were adolescents aged 15–17.9 y. In total, 49.96% (32,168) participants came from urban areas and 50.04% (33,688) came from rural areas. A total of 42.98% participants (27,677) were from the north of China, and 57.02% (36,714) were from the south. Participants from the eastern area, midlands and western areas were 36.32% (23,386), 31.08% (20,015) and 32.60% (20,990), respectively. In total, 6731 (10.45%) children and adolescents were ethnic minorities, and 57,660 (89.55%) were Han ethnicity.

### 3.2. Vitamin D Nutritional Status

The median serum 25(OH)D concentration of Chinese children and adolescents is 17.70 ng/mL (IQR 13.20–22.68 ng/mL) in CNNHS 2016–2017 (Table 1). Boys had significantly higher 25(OH)D concentration than that of girls. The median serum 25(OH)D concentration decreased significantly with age. Children and adolescents live in urban areas had significantly lower median 25(OH)D concentration than those live in rural areas. And the 25(OH)D concentration in children and adolescents from midlands was significantly lower than those from the eastern and western of China. Consistent with our understanding of the impact of latitude and season on the vitamin D status, the 25(OH)D concentration of population in southern areas was significantly higher than that in the northern areas, and it was highest in summer and lowest in winter. Thin children and adolescents had the highest 25(OH)D concentration and obesity ones had the lowest.

The prevalence of vitamin D deficiency, insufficiency and sufficiency of Chinese children and adolescents in CNHHS 2016–2017 was 21.29%, 44.69% and 34.02%, respectively (Table 1). Higher prevalence of inadequacy was significant in girls (71.99%, 25.85% for deficiency and 46.14% for insufficiency) rather than in boys (60.42%, 17.08% for deficiency and 43.34% for insufficiency, *p* < 0.001). The VDD of children (6–11.9 y) was significantly lower than the other age groups. Participants from the midlands of China had significantly higher VDD and VDI than those from eastern areas and western areas. The VDD in northern participants are significantly lower than that in southern ones. Han ethnicity had a significantly higher VDD. In this cross-sectional study, we found that the vitamin D sufficiency of thin participants was highest, then followed by normal participants and finally obesity and overweight ones, and participants with abdominal obesity had a higher VDD and VDI. Moreover, seasonal differences in VDD rates were also significant, with the lowest in summer and significantly increased in autumn, spring and winter in turn. No significant difference was found in different region types in terms of vitamin D inadequacy prevalence.

### 3.3. Risk Factors of Vitamin D Inadequacy

The results of multivariate logistic regression showed that risk factors for vitamin D inadequacy (insufficiency and deficiency are merged together) included (Figure 1): girls had a 1.80 fold increased risk compared to boys (*p* < 0.0001); children aged 12–14.9 y and 15–17.9 y had 2.44 and 2.65 fold of increased risk compared to those aged 6–11.9 y, respectively (*p* < 0.0001); the midlands had a 2.67 fold increased risk compared to the eastern areas; the northern areas had a 3.22 fold increased risk compared to the south (*p* < 0.0001). Spring and winter both increased the risk of vitamin D inadequacy compared to summer (OR = 4.32 for spring, *p* = 0.001; OR = 5.89 for summer, *p* < 0.0001). Compared to normal growth level, overweight and abdominal obesity both increased the risk in relatively low OR values (OR = 1.10 for overweight compared to normal; OR = 1.13 for abdominal obesity compared to those without abdominal obesity). The logistic regression also showed that rural areas (OR = 0.78, *p* = 0.04), minorities (OR = 0.61, *p* = 0.002), thin (OR = 0.87, *p* = 0.001) are protective factors for vitamin D inadequacy. We also performed logistics regressions on the above factors by sex, and the results showed that the risk factors and protective factors were consistent for both sexes.

From multivariate logistic regression, we found that in addition to geographical, seasonal, age and ethnicity factors, growth level and abdominal obesity also affected the nutritional status of vitamin D. Many studies have reported that a higher prevalence of VDI and VDD are related to overweight or obesity in children and adolescents. We also tried to explore whether a similar phenomenon exists in the Chinese population. Significant differences were found in different growth levels in terms of the rate of vitamin D sufficiency, insufficiency and deficiency (Figure 2A). The results showed that overweight children and adolescents had a significantly higher insufficiency rate than normal ones, and thin participants had a significantly higher vitamin D sufficiency than all the other groups. The prevalence of VDD and VDI were significantly higher in abdominal obesity than normal participants. Overall, the overweight population did not have much of an increase in OR value. Abdominal obesity increased the prevalence of VDI and VDD (Figure 2B) and also increased the OR value of VDD in both sexes (Appendix A).

## 4. Discussion

The vitamin D status of Chinese children and adolescents was firstly surveyed in CNHHS 2010–2012 by 25(OH)D radioimmunoassay kits in our laboratory. With the development of detection technology, we have updated the method to LC-MS/MS in the CNHHS 2016–2017. In this new round of the CNHHS 2016–2017, the median serum 25(OH)D concentration was 17.70 (13.20–22.68) ng/mL by the method of LC-MS/MS. The vitamin D insufficiency and deficiency was 44.69% and 21.29%, respectively. We have reported the vitamin D nutritional status of Chinese children and adolescents from CNHHS 2010–2012 [17]. The median serum 25(OH)D concentration was 48.17 (35.36–63.44) nmol/L (19.27 (14.14–25.38) ng/mL) by the method of DiaSorin radioimmunoassay (RIA), and the general prevalence of vitamin D inadequacy was 53.23% (insufficiency 46.00% and deficiency 7.23%) in CNHHS 2010–2012. A direct comparison between the two rounds of data could not be made because the detection methods used were inconsistent. It has been reported that RIA method has a good correlation with the LC-MS/MS method in the determination of 25(OH)D. Ouweland et al. [18] found that DiaSorin RIA was well consistent with the LC-MS/MS method (r^2^ = 0.90, average bias = 1.61 nmol/L) under the vitamin D external quality assessment scheme (DEQAS). Our laboratory also compared DiaSorin RIA method with the LC-MS/MS method. The correlation coefficient of DiaSorin RIA and LC-MS/MS was 0.82, indicating that Diasorin RIA was consistent with LC-MS/MS in the evaluation of VDD and VDI [19], and the fitting equation was Conc.RIA = 0.427 + 1.014 × Conc.LC-MS/MS. To make the comparison, we recalculated the data of CNNHS 2010–2012 by the fitting equations obtained from our previous studies [19]. The converted median serum concentration of children and adolescents of CNNHS 2010–2012 was 18.58 (13.53–20.60) ng/mL. The vitamin D inadequacy was 56.73% (insufficiency 39.37% and deficiency 17.36%). Then, we made the comparison of vitamin D status between CNHHS 2016–2017 and CNHHS 2010–2012. We found that 25(OH)D level decreased and the deficiency prevalence increased significantly in children and adolescents in CNNHS 2016–2017. The same trend of significant increase was also found in pregnant women during the two rounds of CNNHS surveys in terms of VDD rates [20]. The results of the above two rounds of nutrition surveys show that the vitamin D nutritional status in China did not reach the global goal of VDD prevention, which is increasing the serum 25(OH)D level to more than 20 ng/mL in all countries all year long [6]. What is worse, the 25(OH)D level showed a decreasing tendency. Park et al. also reported a seven-year trend of increasing VDD in South Korean adults, according to the Korea National Health and Nutrition Examination Surveys (KNHANES) IV (2007–2009), KNHANES V (2010–2012), and KNHANES VI (2013–2015) [21]. The decreasing 25(OH)D levels and the increasing trend in VDD rates may be partly related to factors, such as lifestyle changes (e.g., reduced outdoor activities) and environmental degradation (e.g., haze).

VDD has become one of the most common health problems in modern society, especially in Asian countries [6]. Even in tropical countries, such as India, VDD is as high as 70 to 100 percent in apparently healthy people due to several socioeconomic and cultural constraints, despite abundant sunshine [22]. The prevalence of VDD (<12 ng/mL) was 26.2%, while that of VDI (12–20 ng/mL) was 37% in Indian children and adolescents during 2016–2017 [23]. On the other side of the world, the vitamin D nutritional status in the United States, Canada and other western countries has been reported less in recent years, but the overall prevalence of VDD is lower than that in Asian countries, and the serum 25(OH)D concentration shows an upward trend. Based on National Health and Nutrition Examination Survey of US (NHANES), modest increases by 5 to 6 nmol/L during 2007 to 2010 was found in terms of serum 25(OH)D levels [24]. The risk of VDD prevalence in the United States remained stable from 2003 to 2014, and the risk of inadequacy declined [25]. The similar trend was also found in Canada on the basis of a ten-year cohort study [26]. In western countries, mandatory or voluntary vitamin D fortification programs have been implemented in line with national policies [27,28]. The fortified foods include milk, cheese, margarine, ready-to-eat cereal, etc. In the US, the average intake of vitamin D from food is approximately 4.9 μg/d, with 2.0 μg/d from natural sources and 2.9 μg/d from fortified foods, which means 60% of dietary vitamin D intake from fortified foods [29]. The fortification policy would partly explain the decrease of VDD in western countries. However, vitamin D fortification is not popular in China or even throughout Asia [22].

Girls have a higher (1.80 fold) increased risk of vitamin D inadequacy than boys in CNNHS 2016–2017, similar to the results found in CNNHS 2010–2012 [17]. The sex difference was very common in both children and adults [30,31]. Less outdoor activity and use of sunscreen for females might be part of the reasons why girls were at a higher risk of vitamin D inadequacy compared to boys. Children aged 12–14.9 y and 15–17.9 y both had more than a 2 fold increased risk of vitamin D inadequacy, possibly due to the gradually increased amount of schoolwork, which takes more time away from outdoor activities. Herrick et al. [25] also reported a higher risk of VDD and VDI in 12–19 y adolescents than in 6–11 y children in NHANES 2011–2014 using the same cutoffs. Children living in rural areas are at less risk for vitamin D inadequacy than those living in urban areas. Air pollution can reduce the level of UVB in the environment, thereby affecting the vitamin D status in the population, especially in urban areas [32]. Sunscreen use by children and adolescents, especially in urban areas, protects against melanoma on the skin, but this may be another reason why vitamin D synthesis is reduced in the skin [33]. The significant impact of latitude on vitamin D status was also shown in our study: northern children had a 3.22 fold increased risk of vitamin D inadequacy than those that lived in the south. We also found that VDD rates were significantly higher in the midlands than in the eastern and western regions, which may be related to different altitudes, amounts of sunlight, and other factors [34]. Season has a significant effect on vitamin D nutritional status; vitamin D sufficiency was up to 55.02% in summer in Chinese children and adolescents, then followed by autumn, spring and winter during 2016–2017. In our study, seasonal variation in 25(OH)D levels was the most prominent among all the subgroups; a 5.89 and a 4.32 fold increased risk of vitamin D inadequacy was found in winter and spring, respectively. This seasonal difference has been widely reported and recognized [35,36], and a retrospective cohort study in Japan found that the seasonal change of vitamin D status even starts to emerge at the age of 2 months [37].

Several studies have reported the inverse relationship of BMI and vitamin D concentrations, and it is explained by the reduced bioavailability of vitamin D (both from skin synthesis and dietary sources) due to its sequestration into a larger pool of adipose tissue [23]. In our study, we also found that overweight child and adolescents had a slightly higher risk (OR = 1.10) of vitamin D inadequacy than those with a normal weight, however, we did not find the same result in those with obesity. In addition, we found that a thin state reduced the risk of vitamin D inadequacy (OR = 0.87). Similar results in thin participants and vitamin D status were rarely reported in this field, a few of which pooled normal and thin populations to compare vitamin D levels with overweight and/or obese populations [38]. The focus of the relevant studies was always between overweight/obesity and 25(OH)D concentration/VDD [39,40]. To further study the impact of overweight and obesity on vitamin D status, we also specifically analyzed the relationship between abdominal obesity and VDI and VDD. We found that the prevalence of VDD and VDI in abdominal obesity children and adolescents was significantly higher than for those who were not, and there was a 1.25 fold increased risk of VDD after confounders’ adjustment. We also found that for every 1 cm increase in waist circumference, the risk of VDD and VDI increased by 1.4% (OR: 1.014 (1.012–1.016)). This is consistent with the result of a systematic review by Hajhashemy et al., which showed serum vitamin D level was inversely associated with risk of abdominal obesity in children and adolescents, in a dose-response manner [41].

We acknowledge several limitations. We were unable to estimate the sun exposure levels, dietary sources of vitamin D, and calcium intake levels for each participant, and we were unable to test relevant bone indicators due to the limited amount of blood drawn from children. Moreover, although vitamin D supplement intake was investigated in our questionnaires, too few participants reported taking supplements therefore the intake of supplements was not included in this study. The factors mentioned above will lead to bias. Nevertheless, our study still has obvious advantages. We have adopted scientific sampling methods and strict quality control measures, and included a large population that is nationally representative. Since 2012, we have been paying close attention to and reporting the vitamin D status of children and adolescents in China. Until now, it is still unclear whether the risk of vitamin D deficiency in children will affect vitamin D status in their adulthood and increase susceptibility to disease; therefore, it is a promising research direction.

## 5. Conclusions

In summary, vitamin D deficiency and insufficiency were very common at above 60% among children and adolescents in China, which is worse than it was five years ago by 10%. Most children and adolescents did not get enough sunlight, particularly during spring and winter. We encourage children and adolescents to participate more in outdoor activities to obtain sufficient and effective sunlight. Vitamin D supplement intake is also encouraged under the recommendation of dietary reference intakes of China, especially in the seasons of spring and winter and for those aged above 12 years, from the urban, midlands and northern areas, and with abdominal obesity.

## Figures and Tables

**Figure 1 nutrients-14-04928-f001:**
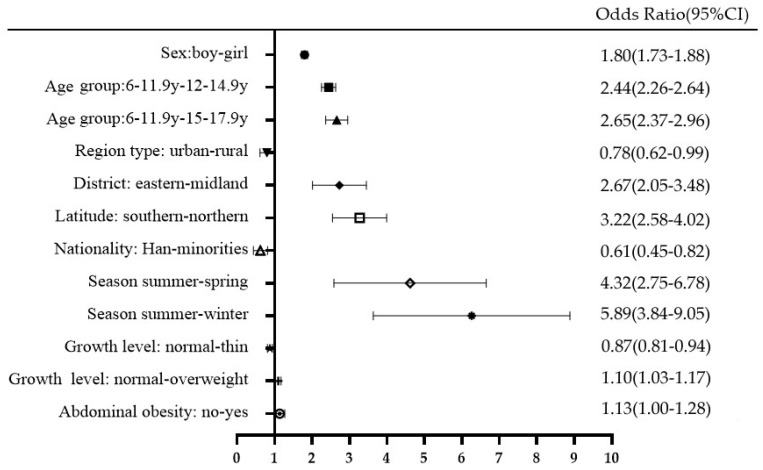
Risk factors for vitamin D inadequacy in CNNHS 2016–2017 (vitamin D insufficiency and deficiency were grouped together as vitamin D inadequacy, 25(OH)D < 20 ng/mL).

**Figure 2 nutrients-14-04928-f002:**
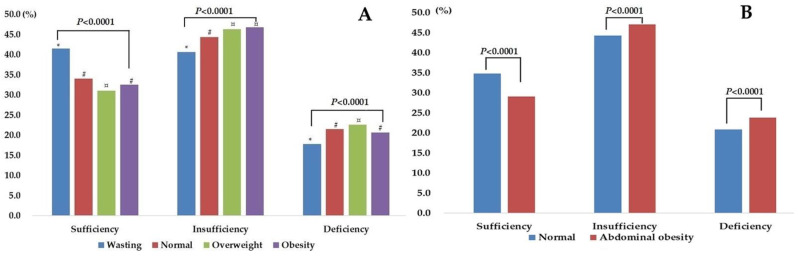
The vitamin D status in different growth level (**A**): *, #, ¤ refers to significant difference between groups; (**B**), normal status versus abdominal obesity.

**Table 1 nutrients-14-04928-t001:** Serum 25(OH)D concentration and vitamin D deficiency for children and adolescents from China National Nutrition and Health Survey of Children and Lactating Mothers in 2016–2017.

Characteristics	No.	25(OH)D Concentration (ng/mL)		Vitamin D Nutritional Status (%, 95% CI)			
		Median (P25–P75)	*p* Value	Sufficient	Insufficient	Deficiency	*p* Value
Total	64,391	17.70 (13.20–22.68)	/	34.02 (31.09–36.94)	44.69 (43.14–46.24)	21.29 (19.00–23.59)	/
Sex			<0.0001				<0.0001
Boys	32,168	18.70 (14.10–23.80)		39.58 (36.43–42.74)	43.34 (41.53–45.15)	17.08 (15.00–19.15)	
Girls	32,223	16.60 (12.40–21.40)		28.01 (25.24–30.76)	46.14 (44.54–47.74)	25.85 (23.23–28.48)	
Age group			<0.0001				<0.0001
6–11.9 y	35,780	19.20 (14.60–24.10) ^c^		43.53 (40.25–46.81)	42.59 (40.57–44.60)	13.88 (12.05–15.72)	
12–14.9 y	15,801	15.90 (11.90–20.70) ^b^		27.11 (24.13–30.08)	46.62 (44.81–48.44)	26.27 (23.43–29.11)	
15–17.9 y	12,810	15.50 (11.60–20.40) ^a^		26.69 (23.63–29.74)	45.89 (43.99–47.79)	27.42 (24.38–30.47)	
Region type			<0.0001				0.51
Urban	30,703	17.40 (13.00–22.30)		32.66 (28.44–36.88)	45.15 (42.94–47.36)	22.19 (18.80–25.59)	
Rural	33,688	17.90 (13.30–23.00)		35.36 (31.32–39.40)	44.24 (42.07–46.40)	20.40 (17.32–23.48)	
District			<0.0001				<0.0001
Eastern	23,386	18.90 (14.10–23.82) ^c^		39.77 (34.87–44.67)	42.61 (40.16–45.06)	17.62 (14.27–20.97)	
Midlands	20,015	15.70 (11.90–20.30) ^a^		22.84 (19.04–26.65)	48.46 (45.90–51.01)	28.70 (24.39–33.02)	
Western	20,990	18.30 (13.70–23.40) ^b^		38.15 (32.72–43.58)	43.47 (40.58–46.35)	18.38 (14.47–22.29)	
Latitude			<0.0001				<0.0001
Northern	27,677	15.12 (11.30–20.00)		22.71 (19.42–25.99)	45.62 (44.02–47.22)	31.67 (28.27–35.08)	
Southern	36,714	19.48 (15.10–24.18)		43.41 (39.44–47.38)	43.92 (41.42–46.42)	12.67 (10.45–14.89)	
Ethnicity			<0.0001				0.0004
Han	57,660	17.48 (13.03–22.40)		32.84 (29.92–35.76)	45.24 (43.70–46.78)	21.92 (19.56–24.28)	
Minorities	6731	19.60 (14.50–24.50)		44.35 (36.98–51.71)	39.86 (35.53–44.20)	15.79 (10.90–20.69)	
Growth level			<0.0001				<0.0001
Thin	6492	19.00 (14.00–24.30) ^c^		41.52 (37.61–45.44)	40.66 (38.18–43.14)	17.82 (15.34–20.30)	
Normal	44,840	17.60 (13.10–22.60) ^b^		34.08 (31.14–37.03)	44.39 (42.82–45.96)	21.53 (19.18–23.88)	
Overweight	7191	17.20 (13.00–21.80) ^a^		31.06 (28.01–34.10)	46.33 (44.52–48.15)	22.61 (20.08–25.14)	
Obesity	5868	17.50 (13.10–22.31) ^b^		32.52 (29.15–35.89)	46.79 (44.72–48.85)	20.69 (17.98–23.41)	
Abdominal obesity			<0.0001				<0.0001
No	58,384	17.80 (13.21–22.80)		34.82 (31.87–37.78)	44.31 (42.74–45.88)	20.87 (18.61–23.12)	
Yes	6007	16.60 (12.50–21.50)		28.85 (25.61–32.08)	47.12 (45.02–49.22)	24.03 (20.94–27.12)	
Season *			<0.0001				<0.0001
Spring	11,545	17.30 (12.90–22.40) ^b^		32.33 (25.95–38.71)	45.14 (41.59–48.70)	22.53 (17.26–27.79)	
Summer	2361	21.70 (17.40–26.20) ^d^		55.02 (46.39–63.64)	38.46 (32.10–44.82)	6.52 (3.24–9.81)	
Autumn	19,223	19.20 (14.80–23.97) ^c^		41.70 (37.39–46.02)	45.20 (42.32–48.07)	13.10 (10.70–15.51)	
Winter	31,262	16.50 (12.19–21.50) ^a^		28.31 (24.39–32.23)	44.68 (42.65–46.72)	27.01 (23.70–30.31)	

* The seasons are divided as follows: March, April and May are spring months; June, July and August are summer months; September, October, November are autumn months; December, January and February are winter months. ^a,b,c,d^ refer to significant differences between groups, different letters indicating a significant difference.

## Data Availability

All data for this article are from CNNHS 2016–2017 and specific data are not publicly available at this time.

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
