# Peer review of "Vitamin D Status for Chinese Children and Adolescents in CNNHS 2016–2017"

_nutrients, 2022, doi:10.3390/nu14224928_

Round 1
Reviewer 1 Report
Thank you for the opportunity to review this interesting article describing the prevalence of vitamin D deficiency (and overall vitamin D status) in children and adolescents in China. This is an interesting and well written article. I only have a few comments to make:
1. The number of decimal points you use in reporting results varies between 2-3 decimal points. I would suggest that you only need to report results to one decimal point, this is then consistent with results of other papers that you compare your work to.
2. At the bottom of table 1 it would be interesting if you could add a key to describe which months of the year are Summer, Winter etc. Readers from around the world will not necessarily be familiar with this.
3. The use of BMI in children - I just wanted to clarify this, you write that BMI was calculated by weight and height, but in the cited article I believe weight and age are used. I presume that age relates to expected height in some respect but a little more detail regarding this would be helpful
thank you
Author Response
Point 1. The number of decimal points you use in reporting results varies between 2-3 decimal points. I would suggest that you only need to report results to one decimal point, this is then consistent with results of other papers that you compare your work to.
Response to Point 1: Thank you for your advice. We have modified the decimal points of OR value into two digits, consisting with the decimal points of 25(OH)D concentration and the rate of vitamin D sufficiency, insufficiency and deficiency. For P values, if there are 2 or more zeros after the decimal point, we retain their original decimal points in order to show the significance of the differences between groups.
Point 2. At the bottom of table 1 it would be interesting if you could add a key to describe which months of the year are Summer, Winter etc. Readers from around the world will not necessarily be familiar with this.
Response to Point 2: Thank you for your advice. We added explanation in Part 2.3 and labeled under table 1 about the division of seasons in terms of months.
Point 3. The use of BMI in children - I just wanted to clarify this, you write that BMI was calculated by weight and height, but in the cited article I believe weight and age are used. I presume that age relates to expected height in some respect but a little more detail regarding this would be helpful
Response to Point 3: Thank you for your advice. Growth levels were classified according to height cut-off points for age and sex and/or BMI of Chinese children and adolescents according to Chinese population standards. We described it in Part 2.3, and we have described a little more about the standard in this revised version.
Reviewer 2 Report
This manuscript mainly studies and analyzes the concentration of 25 (OH) D in children and adolescents aged 6-17 years in China, and evaluates the risk factors for vitamin D deficiency and insufficiency. The 25 (OH) D concentration in the blood of 64391 participants was detected by LC-MS/MS. The participants were classified according to age, sex, region type, ethnicity, season, weight, and height, and 25 (OH) D concentrations in the blood of participants under these groups were analyzed for differences.
Comments:
1. In a manuscript, line numbers appear twice in order from the beginning.
2. In Figure 1, agegroup is missing spaces
3. The authors have described the relationship between regional, seasonal, and abdominal obesity and 25(OH)D. How these factors affect (25(OH)D levels in the blood is not further discussed in the manuscript.
Author Response
Point 1:In a manuscript, line numbers appear twice in order from the beginning.
Response to Point 1: We have renumbered the line numbers of the manuscript.
Point 2. In Figure 1, agegroup is missing spaces
Response to Point 2: We have corrected them in Figure 1.
Point 3. The authors have described the relationship between regional, seasonal, and abdominal obesity and 25(OH)D. How these factors affect (25(OH)D levels in the blood is not further discussed in the manuscript.
Response to Point 3: Thank you for your advice. We have added more discussion about regional, seasonal, and abdominal obesity and 25(OH)D in the 3rd and 4th paragraph of discussion part in revise mode.
Round 2
Reviewer 2 Report
This reviewer accepts the revised version
Author Response
Thank you so much!